# Comparison of the PF07598-Encoded Virulence-Modifying Proteins of *L. interrogans* and *L. borgpetersenii*

**DOI:** 10.3390/tropicalmed8010014

**Published:** 2022-12-26

**Authors:** Dielson S. Vieira, Reetika Chaurasia, Joseph M. Vinetz

**Affiliations:** Section of Infectious Diseases, Department of Internal Medicine, Yale University School of Medicine, New Haven, CT 06510, USA

**Keywords:** Leptospiraceae, pathogenic, protein, vaccination, virulence

## Abstract

Leptospirosis is an emerging infectious disease, with increasing frequency and severity of outbreaks, a changing epidemiology of populations at risk, and the emergence of new strains, serovars, serogroups, and species. Virulence-modifying (VM) proteins encoded by the PF07598 gene family are hypothesized to be *Leptospira*-secreted exotoxins that mediate the molecular and cellular pathogenesis of severe and fatal leptospirosis. If confirmed experimentally, this concept could revolutionize the treatment, diagnosis, prognosis, and vaccine-mediated prevention of leptospirosis by enabling a novel array of targeted interventions. VM proteins, as with other bacterial-secreted protein exotoxins, mediate their virulence effects by attaching to eukaryotic cells, competing with other microorganisms for limited resources in environmental niches, directly intoxicating target cells, and disrupting their function in the mammalian host. In contrast with the most pathogenic group of *Lept ospira*, particularly *L. interrogans*, whose genomes contain 12–15 PF07598 paralogs, strains of the livestock and human pathogen *L. borgpetersenii* have two PF07598 paralogs. Given the possible non-environmentally mediated transmission of some *L. borgpetersenii* strains and the much smaller number of VM proteins in this species, their role in infection and disease may well differ from other leptospiral species. Comparison of VM proteins from different clades of pathogenic *Leptospira* may deepen our understanding of leptospirosis’s pathogenesis, leading to novel approaches to ameliorating *Leptospira* infection in humans and animals.

## 1. Introduction

Human deaths due to infectious diseases remain the leading scourge of humanity [1]. Many of these emerging infectious diseases are zoonotic, originating from both domestic and wild animals [2]. Leptospirosis, a leading yet neglected zoonotic bacterial disease, is responsible for widespread morbidity and disproportionately impacts people in regions with warm, humid climates in which the bacteria can easily proliferate [3]. Difficulties in diagnosing human and animal leptospirosis remain an important problem, and despite the availability of molecular diagnostics, diagnosis still largely depends on archaic serology [4]. The disease is caused by Gram-negative spirochetes of the genus *Leptospira*, which has 64 genomic species and 300 serovars identified to date [5]. A wide variety of animals—rodents, livestock, and wild animals—are sources of leptospiral infection by excreting *Leptospira* in their urine. Humans are accidental hosts and acquire the infection from either direct or indirect sources such as contaminated water, like rivers, puddles, and rice paddies [6,7,8,9,10,11]. Animal leptospirosis is of public health and economic importance as it negatively impairs livestock production and represents an infectious threat to humans; dogs, which both serve as companion animals and roam ferally, are susceptible to leptospiral infection in diverse settings. Cattle leptospirosis can reflect in abortion, stillbirth, premature birth, general reproductive failure, and milk production drop syndrome [12]. Bovines are susceptible to the infection from multiple serovars and *Leptospira* species, including *L. borgpetersenii* serovar Hardjo and *L. interrogans* serovar Pomona [13,14]. Infected animals may be seronegative and still excrete *Leptospira* [13]. The serious health effect on livestock is an important motivation to develop a pan-leptospiral vaccine that would be long-lasting and provide sterile immunity and cross-protection among diverse species and serovars [15]. To achieve a successful vaccination process, it is very important to understand the risk and assessments of the disease in a population.

Fundamental biological insights are essential for understanding how to prevent infectious disease transmission and intervene against disease in both humans and animals. The recently discovered virulence-modifying (VM) proteins, encoded by the PF07598 gene family, are found only in group 1 pathogenic *Leptospira* [16,17,18]. The use of in silico and in vitro approaches have begun to show how these proteins can be an asset in combatting and preventing leptospirosis in humans and animals. This review aims to further understand the potential roles of the PF07598 gene family in the biology of *Leptospira* and the pathobiology of leptospirosis by a comparative analysis of this gene family in *L. interrogans* and *L. borgpetersenii*.

## 2. Leptospirosis and Vaccination

A human being who comes into contact with urine-infected animals or contaminated environments, whether directly or indirectly, is considered at risk for leptospiral infection [19]. Finding a way to prevent and treat this disease is always one of the goals of researchers working with these bacteria. To date, naturally acquired anti-leptospiral immunity has been considered primarily to be antibody-mediated, with immune responses directed against leptospiral lipopolysaccharide (LPS) [20,21,22]. However, it also appears that cell-mediated responses contribute to protection against some serovars such as Hardjo in cattle. In animals, such as dogs, there is currently no gold standard methodology for in vitro prediction of experimental in vivo evaluation of vaccine efficacy [23]. In the case of recombinant vaccines, several vaccine approaches have been tested, including bacterial DNA, viral delivery vaccines, live attenuated bacteria, and subunit vaccines [24]. In the complex extensive reverse vaccinology study to date, a total of 238 where proteins identified and evaluated as potential vaccine candidates [25]. A hamster colonization model was used to evaluate pool of recombinant proteins (5 proteins/pools) and >70% were immunogenic [25]. However, none of the recombinant protein pools conferred protection against kidney colonization [25]. In metanalysis research involving vaccines and vaccination for dogs, researchers estimated a general 84% protection against carrier status [23]. The American Veterinary Medical Association explains that currently available vaccines effectively prevent leptospirosis and protect dogs for at least 12 months [26,27,28]. Annual vaccination is recommended for at-risk dogs and required for bovines. Leptospirosis is a nearly silent infection, rarely inducing outward clinical signs in cattle, as was reflected in peripheral blood mononuclear cell profiles [29].

With few outwardly observable clinical signs, cattle are a chronic host of serologically identical but genetically distinct members of serovar Hardjo, namely *L. borgpetersenii* serovar Hardjo (type Hardjo-bovis) and *L. interrogans* serovar Hardjo (type Hardjo-prajitno) [12]. Cattle are the major reservoir of these agents, which can infect humans and other animals and cause acute disease [30,31]. In North America, *L. borgpetersenii* serovar Hardjo type Hardjo-bovis is most often isolated from cattle [13,31]. Vaccination with bacterins does reduce livestock-human transmission, and it can also reduce the impact of leptospirosis on cattle, which improves animal production [29]. Information is a key feature in the prevention of diseases like leptospirosis. Recently, in research with VM proteins, Chaurasia et al. [32] found that vaccination of C3H/HeJ mice with *L. interrogans* serovar Lai VM proteins protected mice from any clinical manifestations of the disease. The authors observed that this led to ~3–4 log^10^ reduction in bacterial load in the liver and kidney, two key organs in the pathogenesis of leptospirosis and transmission of *Leptospira*, respectively. Hence, recombinant protein vaccines may be key to global immunization against pathogenic leptospirosis, eliminating kidney colonization and spread in the environment [32]. However, so far, we believe this is a very promising hyphothesis.

## 3. Virulence Factors in *Leptospira* and PF07598 Paralogous Gene Family

Virulence in *Leptospira* still has several unknown aspects. Fouts et al. [16] described virulence in terms of survival mechanisms of Leptospira bacteria such as adhesion to the extracellular matrix (ECM), complement evasion and ECM degradation via metalloproteases, motility, chemotaxis, resistance to oxidative stress. As well evading/motility mechanism like the immunodominant proteins of *Leptospira* and the exotoxin PF07598 paralogous gene family. Members of the PF07598 gene family are expressed and upregulated to various extents, as shown in vivo in a hamster model of acute leptospirosis. It has been established that group 1 (pathogenic *Leptospira*) includes *L. interrogans* serovars Icterohaemorrhagiae and Copenhageni, which are the strains most often associated with severe and fatal outcomes [5]. The members of this group uniquely encode in their genomes virulence-related protein families, such as the metalloproteases-associated protein family and the VM protein family [16], which suggests the potential importance of these protein families in the pathogenesis of leptospirosis [5]. An ultimate objective for clinical leptospirosis research is to combat the lack of wide cross-protection provided by current serogroup/serovar-specific bacterin vaccines by developing a pan-leptospirosis vaccine [33,34,35]. The premise of much ongoing research in the field is that identifying pathogenic virulence factors conserved across species/serovar/strain will enable the use of such proteins for vaccine development to target host immune responses that protect against numerous serovars of *Leptospira* [33].

Virulence factors are often the focus of characterization since a bacterin vaccine that exposes an immunized host to more known virulence factors are likely to produce improved protection [33,34,35]. To date, only a limited number of known virulence factors have been identified in pathogenic *Leptospira*, and they have been more intensely studied in *L. *interrogans** than in *L. *borgpetersenii** [16,33]. The outer membrane, surface-exposed virulence factors are of particular focus as they interact directly with the host during infection and are thus available to antibodies [33].

The large novel gene family PF07598, previously of unknown function, encodes the VM proteins found uniquely in group 1 pathogenic *Leptospira* [17,18,36]. The identification of these new virulence-associated genes should spur additional experimental inquiry into their potential roles in mediating *Leptospira* pathogenesis and, perhaps, in enabling *Leptospira* to persist in the hostile and competitive environmental microbial world [17]. Here we take diverse in vitro, in vivo, and in silico approaches to analyze features of the PF07598 gene family at several levels in *Leptospira*.

## 4. Proteins of PF07598 Paralogous Gene Family: General Comparison between Leptospirosis Agents

### Host Tropism

Two of the largest phylogenetically distinct pathogenic species are *Leptospira borgpetersenii* and *Leptospira interrogans*, which, combined, cause most cases of leptospirosis [37]. Although the clinical symptoms of infection due to these two species are similar, they are transmitted differently; epidemiological data support a host-to-host transmission cycle for *L. borgpetersenii* [38]. After the ongoing ecological niche switch from free-living to a symbiotic lifestyle (concomitant with gene expansion), this group of bacteria stabilizes and restricts their lifestyle in specific niches [39]. Overall, so far, is not clear if is the reduction in the number of virulence-modifying proteins in *L. borgpetersenii* associated with a switch to a symbiotic lifestyle that causes protein production changes, although they remain as virulent as *L. interrogans*, which has more VM proteins [12,13,14,15].

In a study comparing *L. borgpetersenii* and *L. interrogans* survival in water at 20 °C, the former lost >90% viability within 48 h whereas *L. Interrogans* retained 100% viability over the same period [38]. *L. interrogans* retained 30% viability over a 3-week incubation, by which time no viable *L. borgpetersenii* were detected (R.L.Z. unpublished data) [38]. The authors concluded that *L. borgpetersenii* does not tolerate nutrient deprivation and does not survive passage through water [38]. *L. borgpetersenii* has a limited capacity to acquire nutrients and survive in environments external to a mammalian host [38]. Although the reason for this is not yet completely clear, it is possible that the large range of potential hosts that can be infected by *Leptospira* species requires some specificity and that horizontal gene transfers may be one of the methods allowing fast adaptation to these hosts [39]. Adherence is only one of the possible mechanisms underlying the host specificity [40]. In some instances, the ability of a pathogen to infect a host has been correlated with its ability to adhere to cells from that species [40]. This leads to more evolutionary questions about the *Leptospira* species and host dynamics.

## 5. General Hypothesis

Our body of work has shown that the PF07598 gene family is restricted to group 1 pathogenic *Leptospira* and is not found in either intermediate or saprophytic *Leptospira* [16,17,18,36,41]. VM proteins encode the paralogs, which vary in their numbers *in L. interrogans* serovar Lai (12), *L. interrogans* serovar Copenhageni (13), *L. borgpetersenii* serovar Hardjo (4), and several others using system biology and “pathogenomic” approaches [16]. This raises several questions. For instance, why does *L. borgpetersenii* have four VM proteins? There may be one protein and three isomers, or there may be four different proteins. Perhaps there have also been mildly deleterious mutations or, similarly, new insertions of intrinsic elements that might facilitate intragenomic recombination.

A tenet of evolution is that alleles that are under negative selection are often deleterious and confer no evolutionary advantage; these are removed from the gene and are eventually extinguished from the population [42]. Conversely, alleles under positive selection do confer an evolutionary advantage and lead to an increase in the overall fitness of the organism; thus, they increase in frequency until they eventually become fixed in the population [42]. Comparing the complete genome of two strains of *L. borgpetersenii* serovar Hardjo, one hypothesis is that the bacteria are undergoing a process of insertion sequence (IS)-mediated genome reduction [38], leading simultaneously to the perception of genome reduction in *L. borgpetersenii* and genome expansion in *L. interrogans*, which reflects differences in the environments during transmission between hosts. That is one theory that could support the evolution of *L. borgpetersenii*. Based on several comparisons [16,17,18,43], the interpretation is that Lai has orthologs in *L. interrogans* Copenhageni (LIC) apart from LIC_10639, and each protein from *L. borgpetersenii* has copies inside the species. Answering the general question about why this happens is a work in progress.

## 6. Amino Acids and Basic Evolutionary Differences

Since LIC and *L. interrogans* Lai cause severe leptospirosis in humans and *L. borgpetersenii* Hardjo-bovis causes fetal loss in cattle, we focused on the phylogenetic comparison between the VM proteins from these species and serovars (Table 1). In the phylogenetic tree, the evolutionary history was inferred using the neighbor-joining method and MEGA 11 [44,45]. The selection from the *L. borgpeterseni* protein sequences was based on a previous result from Putz et al. [43], where the authors demonstrated that the genes Q04NE0, Q04T47, and Q04V07 are present in a clinical isolate from cattle.

*Leptospira borgpetersenii* serovar Hardjo strains HB203 and JB197 have a high level of genetic homology but cause different clinical presentations in the hamster model of infection [43]; HB203 colonizes the kidney and presents with chronic shedding while JB197 causes severe organ failure and mortality [43].

Comparative analysis of protein content suggests that different protein profiles are expressed by *Leptospira* strain JB197 compared to strain HB203 (both cultivated at 29 and 37 °C) [43]. For the genome of JB197, the proteins Q04T47 and Q04V07 were highly expressed (proteomics analysis) compared with Q04NE0, but with a low expression still present in the genome. In UniProt Reference Clusters using sequence identity thresholds (100%, 90%, and 50%), we determined that M6BFG1 from *L. borgpetersenii* serovar Hardjo-bovis strain Sponselee showed 90% identity with Q04V07. Based on this finding, *L. borgpetersenii* serovar Hardjo-bovis presents four VM proteins, and further experimental validations could be used to develop a potential vaccine candidate for cattle and other animals worldwide.

As expected, in the phylogenetic tree, *L. borgpetersenii* serovar Hardjo-bovis M6BFG1 was clustered with Q04V07 with a bootstrap value of 100%, and with Q04NE0 and Q04T47 with a bootstrap value of 99% (Figure 1). Sequence data of protein from multiple species help in understanding evolutionary patterns. Multiple Sequence Alignment (MSA) of protein sequences obtained from across species or a cohort is usually the starting point for such analyses [46].

Comparative multiple sequence analysis of VM proteins from *L. interrogans* Lai (ancient VM protein sequences LA1400 and LA1402, vaccine candidates) and *L. borgpeterseni* serovar Hardjo-bovis (Figure 2) showed non-conserved amino acids specific to Lai and Hardjo-bovis. This clearly differentiates the significance of species-specific amino acids’ variation-conservation and their functional role in the pathogenesis of human and animal leptospirosis. Conservation of amino acids at the N-terminal and variation at the C-terminal revealed that these VM proteins can present different degrees of host tropism. However, more research must be performed to further investigate that aspect.

When evaluating the level of conservation of individual amino acids [46], a highly conserved amino acid is likely to have an important role in either structure or function. This is especially true for perfectly conserved amino acids, which are mostly identified in the functional sites of proteins [46]. Focusing on the differences between proteins in the same gene family can lead to future studies about evolutionary aspects beyond cross-reaction in vaccine research. This suggests that the amino acid specific to Hardjo-bovis and non-conserved on Lai could be a potential target for animal vaccines and therapeutics.

According to Anfinsen’s dogma of molecular biology, the native structure and function of proteins are uniquely determined by their amino acid sequence [48]. To identify the amino acid composition, the Expasy ProtParam [49] tool was used to create figures that would highlight the differences between *L. borgpeterseni* and *L. interrogans* (Figure 3), thus capturing the importance of understanding a phylogenetic analysis.

Knowledge of the complete amino acid sequence of each protein (Figure 3) is crucial not only to the basic understanding of its molecular structure, but also for defining the correlation of the protein structure with the biological and immunologic properties of, say, bacteria [50]. Understanding VM proteins helps develop one strong final product since amino acid deletion can strongly affect protein function. Based on the similarity and homology of sequences curated from different species, protein sequences are classified into families that are likely to share structural and functional similarities [46]. Most sites tolerate a degree of change, which reflects either a polymorphism or a mutation under a selection pressure [46]. However, co-evolution, where another amino acid simultaneously undergoes a mutation along with it, has recently gained considerable interest [46]. The correlation between the phylogenetic tree (Figure 1) and the content of the amino acid (Figure 3) represents a range of data that can give a background for more in vitro and in vivo analysis of these proteins. Besides that, it is worthwhile to consider which protein characteristics are important for which processes or functions.

## 7. Antigenicity and Epitopes

Vaccination, which involves several natural and artificial proteins, is a key strategy for controlling various infectious diseases. This gives another layer to understanding antigenicity and epitopes, as well as their importance in the protein structure. Computational vaccine design, also known as computational vaccinology, encompasses epitope mapping (Figure 4), antigenicity (Table 2), and immunogen design tools. In silico prediction of immune response to emerging infectious diseases can accelerate the next generation of vaccines. The capacity of an antigen to bind with the receptors is called the antigenicity [47]. Characterizing the antigenicity of a protein can be important for predicting vaccine efficacy.

However, such experiments are labor-intensive, time-consuming, and not suitable for the early stage detection [47]. Computational prediction of antigenic dissimilarity using amino acid sequences enables large-scale antigenic characterization [51]. This approach for example can be applied as well, to reverse vaccinology (RV), which is a technique that has been widely used for screening surface-exposed proteins (PSEs) of important pathogens, including outer membrane proteins (OMPs), and extracellular proteins (ECPs) as potential vaccine candidates [52]. One of the goals of using this method is the initial screening is achieved by using bioinformatics to identify all surface-exposed proteins (potential vaccine candidates) and this typically reduces the number of targets 10-fold, from thousands to hundreds of proteins [51,53]. Surface-exposed proteins generally comprise a wide array of virulence factors involved in pathogen-host interactions and are responsible for causing disease [52]. Overall, Screening using in vitro assays further reduces the number of vaccine candidates and hence the number of laboratory animals required for efficacy testing.

Importantly, sequence-based computational methods make it possible to characterize antigenicity without requiring high biosafety levels [47]. The use of programs such as VaxiJen showcases the idea that antigens possess common, underlying physicochemical features that are independent of species and conventional global sequence similarity [54]. The VM proteins presented here, based on our results, can be recognized by the binding sites of the antibodies.

With the main objective of producing vaccines for humans and animals, after understanding that the protein can be recognized by antibodies, the next step is to understand the localization of epitopes inside the amino acid sequences. A B-cell epitope is the antigen portion binding to the immunoglobulin or antibody [55]. These epitopes recognized by B-cells may constitute any solvent-exposed region in the antigen and can be of different chemical natures and can be subjects for epitope prediction methods [55]. B-cell epitope prediction aims to identify B-cell epitopes with the practical purpose of replacing the antigen for antibody production or for carrying structure-function studies [55]. Any solvent-exposed region in the antigen can be subject to recognition by antibodies [55].

These data support the next steps for protein in vivo research (Figure 4 and Table 2). VM proteins are members of the PF07598 family, which is gaining attention as studies lead to potential human and animal vaccination. Based on what was previously explained about *Leptospira* vaccines, cross-reaction between vaccines from a protein produced by different species is very relevant to the field. This leads to challenges that need to be overcome.

## 8. Clinical Relevance and Challenges

The highly innovative aspect of the present body of work is the discovery of VM proteins and the hypothesis that these molecules are possible exotoxins that mediate severe and fatal leptospirosis at the cellular level. Despite years of study and the availability of commercial vaccines, detailed analysis of the animal immune response to vaccination and the *Leptospira* challenge is lacking [29]. This remains a problem even in the face of technology related to vaccine production. Vaccines for *Leptospira* with cross-reaction potential and strong protective aspects are still not available for humans and animals. Not just that, but there are several serovars and strains related to *L. borgpetersenii* (Table 3). The use of VM proteins could help close this gap or at least be a starting point.

In addition, overall surveillance should prioritize species where economic losses have been identified, species with a known history of maintaining the occurrence of leptospirosis, and novel species identified in epidemiological studies of the human disease [12]. This will establish the relative importance of serovars and host species, leading to possibilities for reducing/managing risk factors, vaccination programs, rodent control programs, and animal health programs [12].

Therefore, vaccine development would be a critical, and possibly the first, demonstration of the importance of conserved proteins for the vaccine at the species level, which would represent a major advancement in human and animal leptospirosis vaccine and adjunct biologics development.

## 9. Concluding Remarks

The unique “lifestyle” of *Leptospira* species has given a different direction to their genome features. Virulence-modifying (VM) proteins are still a novel approach to discovering more about leptospirosis pathogenesis, presenting an opportunity for new treatments and prevention protocols. The use of in silico and in vitro approaches have started to show how these proteins can be an asset in combatting and preventing leptospirosis in humans and, in the future, animals. While previous mouse immunization experiments suggest that VM proteins are important virulence factors in vivo, the proposed mechanism of protective immunity in the animal system remains unknown. Demonstrating that immunization with VM proteins from one *Leptospira* species and serogroup cross-protects against challenge infection containing a different serogroup would be an important advancement in the field. Such research would form the basis for leptospirosis vaccine development for several species.

## Figures and Tables

**Figure 1 tropicalmed-08-00014-f001:**
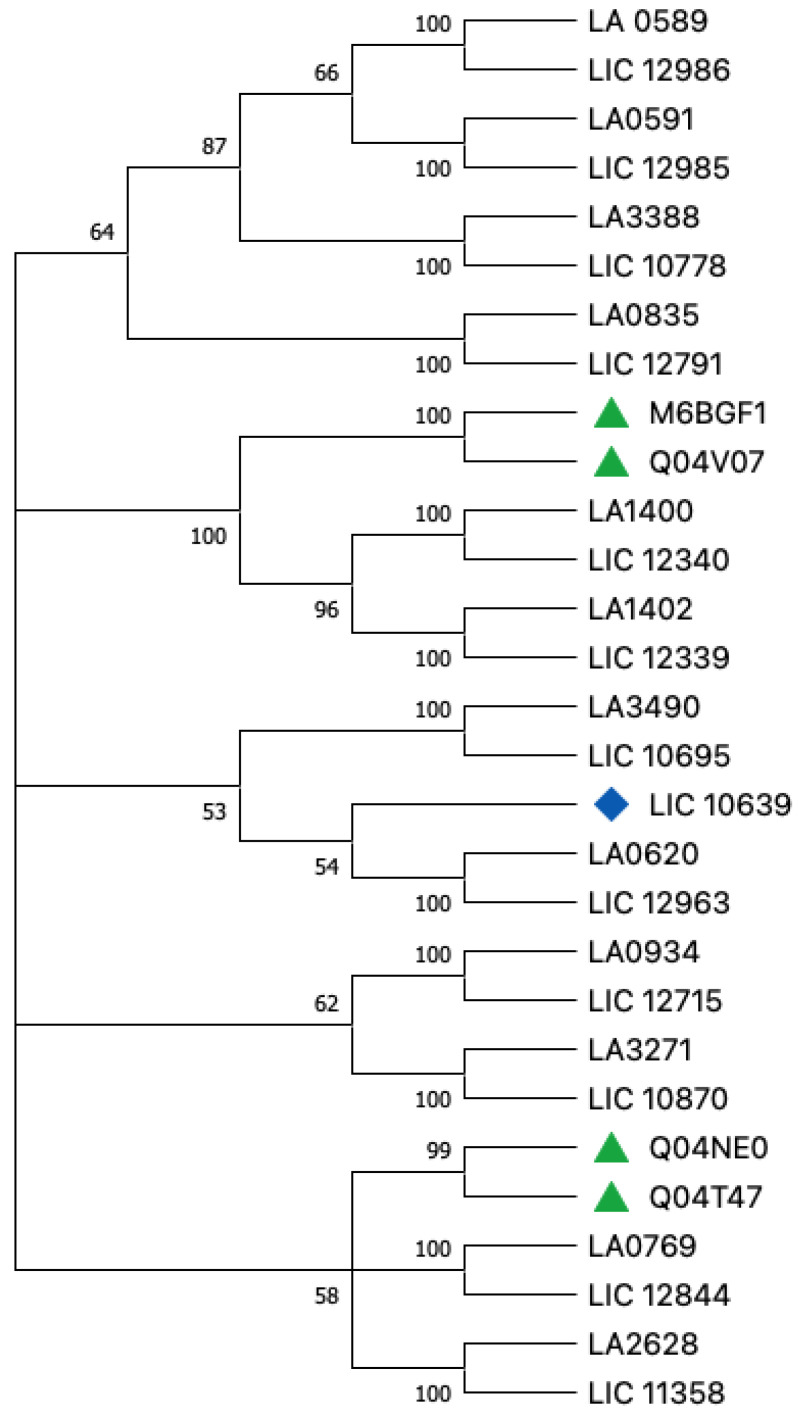
Preliminary phylogenetic tree using amino acid sequences from VM proteins from *L. interrogans* Lai (LA), *L. interrogans* Copenhageni (LIC), and *L. borgpeterseni*, based on Fouts et al. [17] and Putz et al. [37], data. Legend: LA is related with *L. interrogans* Lai; Green (triangle)—*L. borgpeteersenii* VM proteins; Blue (rhombus)—LIC is the VM protein that does not have an ortholog as a match and is related with *L. interrogans* Copenhageni.

**Figure 2 tropicalmed-08-00014-f002:**
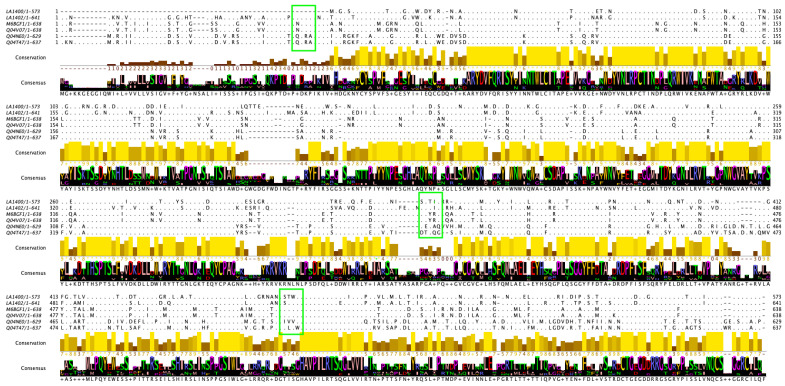
Multiple alignments for consensus, conservation, and direct differences were performed with Jalview [47] using amino acid sequences of *L. interrogans* Lai and *L. borgpeterseni* VM proteins. Legend: The green box shows highlights os similarities and differences in the amino acids of different strains of *Leptospira.*

**Figure 3 tropicalmed-08-00014-f003:**
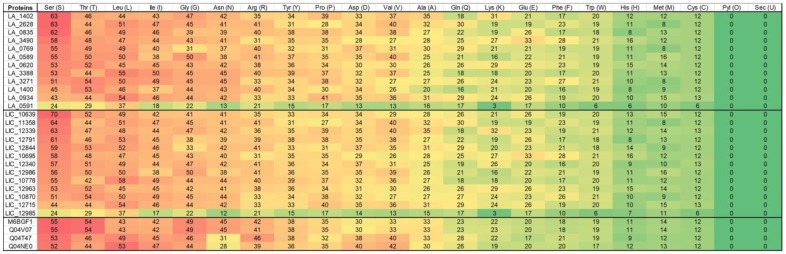
Comparative amino acids conservation from *L. interrogans* Lai, *L. interrogans* Copenhageni (LIC), and *L. borgpeterseni*. The prediction of amino acid conservation across the spp. among the paralogs was performed using Expasy ProtParam tool [49]. Legend: Redder, the bigger the amount of that amino acid at the VM protein sequence. Greener, less the amount of that particular amino acid at the VM protein sequence.

**Figure 4 tropicalmed-08-00014-f004:**
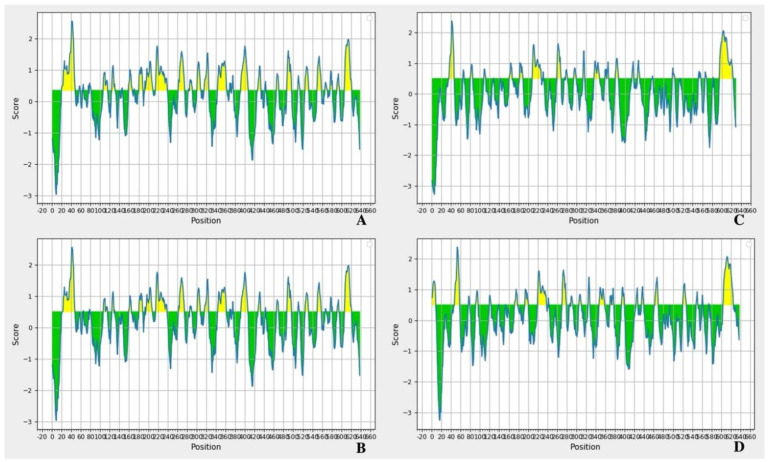
Epitope modeling using amino acids sequences of *L. borgpeterseni* VM proteins applied at the Bepipred Linear Epitope Prediction tool [51]. Legend: (**A**) (M6BGF1), (**B**) (Q04V07), (**C**) (Q04NE0) and (**D**) (Q04T47). Better visualization of the figures can be found at the Appendix A as well as the figures related to *L. interrogans* LAI epitope modeling.

**Table 1 tropicalmed-08-00014-t001:** General List of virulence-modifying proteins from *L. interrogans* Lai, *L. interrogans* Copenhageni and *L. borgpeterseni*, gene code, amino acids amount, and signal peptide prediction using Signal P 6.0 server, all based on Fouts et al. [16] and Putz et al. [43] data.

*Lepstospira* Strain	Gene	Protein ID	Amino Acids	Cleavage Site
***L. interrogans* Lai**	LA_3490	Q8F0K3	639	18aa–19aa
	LA_0620	Q8F8D7	637	29aa–30aa
	LA_1402	Q8F6A7	641	27aa–28aa
	LA_1400	Q8F6A9	573	NA
	LA_0591	Q8F8G6	313	22aa–23aa
	LA_3388	Q8F0V3	631	22aa–23aa
	LA_0835	Q8F7V7	631	22aa–23aa
	LA_0589	Q8F8G8	632	22aa–23aa
	LA_3271	Q8F166	636	31aa–32aa
	LA_0934	Q8F7L0	638	31aa–32aa,
	LA_0769	Q8F820	602	NA
	LA_2628	Q8F2Y3	638	22aa–23aa
** *L. borgpeterseni* **	LBJ_1339	Q04T47	637	30aa–31aa
	LBJ_0577	Q04V07	638	22aa–23aa
	LBJ_4195	Q04NE0	629	19aa–20aa
	LEP1GSC016_2732	M6BGF1	638	22aa–23aa
***L. interrogans* Copenhageni**	LIC_12986	Q72N52	632	22aa–23aa
	LIC_12339	Q72PX8	663	NA
	LIC_12340	Q72PX7	627	30aa–31aa
	LIC_10695	Q72UG2	639	30aa–31aa
	LIC_10639	Q72UL8	640	33aa–34aa
	LIC_12963	Q72N74	637	31aa–32aa
	LIC_12844	Q72NJ0	639	24aa–25aa
	LIC_10870	Q72TZ4	636	31aa–32aa
	LIC_12715	Q72NW3	638	31aa–32aa
	LIC_11358	72SM1	638	22aa–23aa
	LIC_12791	Q72NP1	631	22aa–23aa
	LIC_10778	Q72U83	631	22aa–23aa
	LIC_12985	Q72U83	631	22aa–23aa

**Table 2 tropicalmed-08-00014-t002:** Antigenicity prediction using amino acids sequences from *L. interrogans* Lai and *L. borgpeterseni* VM proteins, using VaxiJen [29] server.

Protein	Antigenicity	Prediction for the Protective Antigen
LA1400	+	0.4561
LA1402	+	0.3814
M6BGE1	+	0.3925
Q04V07	+	0.3998
Q04NE0	+	0.3928
Q04T47	+	0.4243

**Table 3 tropicalmed-08-00014-t003:** List of taxon ID (NCBI) of *L. borgpetersenii*, with different serovars.

Taxon Id	Scientific Name
1303729 *	*Leptospira borgpetersenii* serovar Hardjo-bovis str. Sponselee
280504	*Leptospira borgpetersenii* serovar Javanica
280505	*Leptospira borgpetersenii* serovar Ballum
280506	*Leptospira borgpetersenii* serovar Tarassovi
280507	*Leptospira borgpetersenii* serovar Mini
328971	*Leptospira borgpetersenii* serovar Hardjo
338217	*Leptospira borgpetersenii* serovar Hardjo-bovis
338220	*Leptospira borgpetersenii* serovar Sejroe
355277 *	*Leptospira borgpetersenii* serovar Hardjo-bovis str. JB197
376921	*Leptospira borgpetersenii* serovar Castellonis
400680	*Leptospira borgpetersenii* serovar Saxkoebing
400681	*Leptospira borgpetersenii* serovar Wolfii
508535	*Leptospira borgpetersenii* serovar Balcanica
508536	*Leptospira borgpetersenii* serovar Ceylonica
508537	*Leptospira borgpetersenii* serovar Dikkeni
508538	*Leptospira borgpetersenii* serovar Jules
508539	*Leptospira borgpetersenii* serovar Kisuba
508540	*Leptospira borgpetersenii* serovar Kwale
508541	*Leptospira borgpetersenii* serovar Moldaviae
508542	*Leptospira borgpetersenii* serovar Nero
508543	*Leptospira borgpetersenii* serovar Nyanza
508544	*Leptospira borgpetersenii* serovar Pina
508545	*Leptospira borgpetersenii* serovar Poi
508546	*Leptospira borgpetersenii* serovar Polonica
508547	*Leptospira borgpetersenii* serovar Sorexjalna
508548	*Leptospira borgpetersenii* serovar Tunis
508549	*Leptospira borgpetersenii* serovar Worsfoldi
508550	*Leptospira borgpetersenii* serovar Srebarna
561995	*Leptospira borgpetersenii* serovar Whitticombi
577404	*Leptospira borgpetersenii* serovar Istrica
652580	*Leptospira borgpetersenii* serovar Arborea
1049771	*Leptospira borgpetersenii* serovar Kenya
1049780	*Leptospira borgpetersenii* str. Brem 328
1049781	*Leptospira borgpetersenii* str. Noumea 25
1049782	*Leptospira borgpetersenii* str. UI 09149
1049786	*Leptospira borgpetersenii* str. Brem 307
1141102	*Leptospira borgpetersenii* str. 4E
1192865	*Leptospira borgpetersenii* serovar Pomona
1193007	*Leptospira borgpetersenii* str. 200701203
1193008	*Leptospira borgpetersenii* str. 200801773
1193009	*Leptospira borgpetersenii* str. 200801926
2034701	*Leptospira borgpetersenii* serovar Piyasena
2814707	*Leptospira borgpetersenii* serovar Guangdong
2814708	*Leptospira borgpetersenii* serovar Hamptoni
2814709	*Leptospira borgpetersenii* serovar Nigeria

Observation: The list from UNIPROT (https://www.uniprot.org/taxonomy?query=parent:174; accessed on 8 November 2022 ) contains 43 “children” from *L. borgpetersenii*, two strains that have been used in applied research were added to the list and are marked (*). In the same link is possible to find taxon ID that presents proteomics data.

## Data Availability

The datasets presented in this study can be obtained from the authors when resquested.

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
