# Peer review of "Comparison of the PF07598-Encoded Virulence-Modifying Proteins of L. interrogans and L. borgpetersenii"

_tropicalmed, 2022, doi:10.3390/tropicalmed8010014_

Round 1

Reviewer 1 Report

Editorial (textual) comments:

1. Page 1 line 17: replace "vaccine prevention" with "vaccine-mediated prevention".

2. Page 2 lines 79-80: replace "pool" with "pools".

3. Page 3 line 101: reference 27 is the same as reference 16, therefore replace 27 with 16. In addition, correct the reference numbers under References, page 12-15.

4. Page 3 line 106: after "survival mechanisms" add: "of Leptospira bacteria".

5. Page 4 lines107-109: "immunodominant proteins" and "the PF07598 paralogous gene family" are not mechanisms, whereas all other terms in this sentence are examples of mechanisms. Please make clear what is meant with these two terms; what mechanism.

6. Page 4 line 114: Please keep consistency in the spelling of metalloprotease or metalloproteinase. Both is correct but just choose one spelling for this throughout the paper.

7. Page 5 line 200: In this text there is no mentioning of the serovar(s) of strains JB197 and HB203, only that they are Leptospira. Also they are not listed in Table 1 so that their serovar designation cannot be read in this Table either. Please add the serovar(s) here and, if relevant, the Leptospira species.

8. Page 7 lines 239-241, Legend of Figure 3: reference(s) is missing.

9. References: see editorial comment no. 3 above. In addition, also ref. 18 is identical to ref. 28; so please correct this as well.

Substantive/general comments:

This review clearly describes the backgrounds of, insights into and importance of the recently discovered virulence-modifying proteins that are found in the more pathogenic group of Leptospira species. In particular the link of the probable function of these proteins with application in the field of vaccine development is highlighted, including the potential to induce cross-Leptospira species and cross-Leptospira serogroup protection, which is not possible with the currently available "bacterin" vaccines.

However, I am wondering if any passive immunization studies (e.g. using the hamster infection model) with polyclonal or monoclonal antibodies against certain VM proteins have been done and if so, why these studies are not included in this review. Apart from relevant in vitro and ex vivo studies on the possible function and antigenicity of VM proteins, it is highly significant to report examples of active immunization and protection (included in review, ref. 16) as well as examples of passive immunization and protection, which are not included.

Since in older Leptospira literature several examples are given of the induction of the more virulent in vivo phenotype of Leptospira through upregulation of certain outer membrane (lipo)proteins by mimicking the in vivo conditions during in vitro culturing (e.g. temp/osmolarity), I am wondering if this has also been demonstrated for the production and secretion of VM proteins. If so, it would be recommended to include this in this review. 

Author Response

see attached please

Reviewer 2 Report

This paper is very well written and reviews pathogenic Leptospira virulence modifying proteins (VM) discovered in the lab of Dr. Vinetz. It is an important contribution to the field of leptospirosis.

Overall: For a review the paper lacks citations of original research to substantiate the claims; it also lacks references to other groups contributions. For example, lines 66-71, 2 references are cited that are nearly irrelevant to the statements made. Ref 20 should be an epidemiological review and ref 21 should be original research on antibody responses to Lepto LPS. The paper reads like a re-referencing of other reviews as well as the work done by the PI lab. This review will have much more value when the relevant original research references are added. Other groups should be acknowledged for their contribution in common areas of expertise.

Specific comments:

Lines 59-64: you need a stronger rationale for comparing these two species since both are equally highly pathogenic & produce similar clinical outcomes and L. interrogans has 12-15 VM PF07598 paralogs but L. borgpetersenii has 4. A better explanation is given in lines 140-146.  Is the reduction in number of virulence modifying proteins in L. borgpetersenii associated with a switch to symbiotic lifestyle although they remain as virulent as L. interrogans that has more VMs?

Line 73, Ref 22 is in dogs, so be specific.

Lines 101-102: this is purely speculative, there is no evidence that this can be done.

Lines 116-119: how does ref 29 support this claim, as well as the claim in lines 123-125?

Figure 1, LA stands for L interrogans Lai? If so specify in legend

Figure 3 needs better captions and figure legends. What does red mean?

In Table 3, since you are comparing L. interrogans and L. borgpetersenii I wonder if you should also include a table listing all the L. interrogans taxon ID with different serovars (for quick reference).
